# Adverse childhood experiences and elder abuse victimization nexus: A systematic review and meta-analysis

Kofi Awuviry-Newton[1,2]*, Bernadette Saunders[1], Nestor Asiamah[3], Kwamina Abekah-Carter[2,4], Daniel Doh[2,5]

1 Department of Allied Health, College of Sport, Health and Engineering, Victoria University, Melbourne, Australia, 2 African Health and Ageing Research Centre, Winneba, Central Region, Ghana, 3 School of Health and Social Care, University of Essex, Colchester, United Kingdom, 4 Memorial University of Newfoundland, St. John's, Canada, 5 School of Allied Health, The University of Western Australia, Perth, Australia

* Kofi.Awuviry-Newton@vu.edu.au

**Data Availability Statement:** All relevant data are within the paper. This paper is a systematic review and meta-analysis so primary papers used are all in the paper.

## Abstract

Adverse childhood experiences (ACEs) are important life course events that can influence elder abuse victimisation (EAV) among older adults. This systematic review and meta-analysis aimed to provide synthesised and consolidated evidence on the existing associations between ACEs and EAV. A systematic search was conducted across six databases, including PubMed, PsycINFO, CINAHL Complete, Scopus, Google Scholar, and the Web of Science. All studies that addressed associations between ACEs, in singular or multiple form, and EAV were included in the review. Meta-analysis of the extracted odds ratios (ORs) and confidence intervals (CIs) was conducted using the common-effect inverse-variance model. Nine studies (cross-sectional design = 7; cohort design = 2) met the inclusion criteria. Included studies examined multiple ACEs and multiple EAVs associations (N = 3); at least single ACE and multiple EAVs (N = 3); any single form of ACE and multiple EAVs (N = 3); multiple ACEs–any single form of EAV nexus (N = 2); multiple ACEs–financial elder abuse association (N = 2); and multiple ACEs–physical elder abuse nexus (N = 2). Pooled ORs and CIs showed statistically significant results for all ACEs and EAVs associations whether in singular or multiple form. The results indicate that interventions designed to reduce ACEs, in singular or multiple form, early in life targeting residential and community-dwelling older adults may be relevant in reducing the incidence of EAV. The life course perspective s be integrated into the planning for support services for children, families, and older adults to prevent EAV in singular or multiple forms in later life.

## Background

The growing prevalence and incidence of elder abuse victimization (EAV) reported in residential and community (home care) settings across the world poses both a public health and a human rights concern. A systematic review of elder abuse prevalence in community settings

**Funding:** The author(s) received no specific funding for this work.

**Competing interests:** The authors have declared that no competing interests exist

revealed that 1 out of 6 older adults (aged 60 years and above) report experiencing abuse of various forms [1]. For older adults living in residential and long-term care facilities, the prevalence and experience are worse with evidence reporting that two-thirds of aged care staff admit committing abuse in the past year [2]. In addition to the reported increase in abuse during the Coronavirus disease– 2019 (COVID-19) [3], the prevalence of abuse in many countries, including low- and middle-income countries, is expected to increase with the increasing rate of population ageing [1, 4, 5]. Increasing experiences of elder abuse are associated with adverse childhood experiences (ACEs) such as high levels of childhood maltreatment [6].

EAV, in this review, will refer to older adults' experiences of a single or repeated act or lack of appropriate action, occurring within any trusting relationship that causes harm or distress to older adults in residential or community settings [7]. Examples of this victimisation include older adults' experiences of ageism, financial exploitation, sexual abuse, physical and psychological assault or abuse, and neglect [8]. Recently, available studies that employed a life course perspective revealed that EAV in later life may be influenced by childhood experiences of adverse circumstances including abuse [9, 10]. However, to our knowledge, no systematic syntheses or reviews currently exist to explore the nature of this connection/pathway. ACEs refer to any direct harm resulting from the childhood experience of abuse and neglect as well as the indirect harm emanating from any negative family or social environments [11, 12]. Specific examples of ACEs include childhood sexual abuse, parental criminality, parental substance use, and violence against the mother [13]. In addition to the original list of ACEs [13], more recent research has offered broadened definitions that include childhood physical abuse, family conflict, exposure to family violence, family financial problems, childhood discrimination, and the cumulative harmful effects of at least two domains of ACEs [11]. We contend that childhood adversities or abuse may have a connection with EAV through several pathways. One of the pathways, explored in this study, is that older adults who experienced adverse circumstances in their childhood, living in any setting, may carry disempowerment with them throughout life, and are more likely to be re-victimized in the community or a residential setting [14]. The connections between ACEs and EAV are relevant in developing early interventions to reduce the incidence of abuse in later life.

Studies on the associations between ACEs and adverse outcomes vary in their designs, methodologies, locations, findings, and conclusions. For example, a study conducted in China revealed that social and economic instability during childhood influence an increase in the prevalence of poor health outcomes, such as obesity, difficulty with daily living, and exhibiting depressive symptoms [15]. Further, many systematic reviews explore ACEs and varying adverse outcomes, but not EAV. For instance, systematic reviews have provided evidence on ACEs, or effects of multiple ACEs, on health outcomes such as sexual, mental, and physical health, and higher utilisation of healthcare [11, 16, 17]. In addition to an earlier study [18], more recent research has focused on the association between ACEs and EAV [9, 19, 20]. For instance, in the United States (US), a longitudinal study showed that childhood adversities can have a significant direct effect on EAV [9]. However, what is lacking is synthesized evidence on the ACEs–EAV association. Synthesizing evidence on this association will contribute to understanding, or revealing, a clear pathway of the association and evidence on how to mitigate this public health concern.

As applied elsewhere [18], we employed the life-course perspective as a useful framework for interpreting the findings of the systematic review and presenting evidence for policies, research, and practice about aspects of developmental experiences that need to be tackled to reduce the incidence of EAV in later life. According to Elder [21], the life-course perspective elucidates the potential effects of accumulative advantages or disadvantages on individuals over time resulting from their relationship with social structures. The framework posits that

early developmental experiences–in life stages such as early childhood, adolescence, young adulthood, middle age, and late adulthood–influence later life experiences [21, 22].

This systematic review aimed to understand the connections between ACEs and EAVs, as well as gain insight into factors moderating the relationships between ACEs and EAVs. An enhanced understanding of this evidence using a life-course perspective would assist in the development of appropriate interventions for older adults who experienced single or multiple ACEs, prevent the occurrence of ACEs, and improve cross-cultural and longitudinal research on the relationships between ACEs and EAVs. The understanding from this systematic review will strengthen the existing practice approaches, support services, and policies available to older adults experiencing abuse in old age and children experiencing ACEs of various forms.

## Methods

### Protocol

The Preferred Reporting Items for Systematic Reviews and Meta-Analysis (PRISMA) checklist [23] was used to guide the systematic review and meta-analysis (see S1 File). The protocol of the current review has been submitted to PROSPERO, with a registration number CRD42023452985.

### Study eligibility criteria

Two authors (KAN and KAC) did the screening independently and met on Zoom to address any disagreement and achieve consensus. Two authors (BS and NA) were brought in to review the papers for eligibility and address any disagreements that were not settled by KAN and KAC. All studies that explored the connection between ACEs and EAV and satisfied the following criteria were included in the systematic review:

a. Older adults (60years and older) as the main study participants

b. Studies measuring ACEs in general or any of the domains and associations with EAV. The domains of ACEs include childhood factors and parental factors (see Table 1 for details).

c. Studies published in English in refereed peer-reviewed journals.

d. Studies on EAV including ageism, psychological abuse, physical abuse, financial abuse, neglect, or sexual abuse (see Table 1 for details).

Study inclusion was not restricted to any research approach, year of publication, location of study, or type of analytical method used to examine the association between ACEs and EAV. These inclusion strategies allowed for any study available on the ACEs–EAV association to be included in the systematic review. Articles identified in the databases searched were exported to Endnote, a reference manager, to allow for duplicate removal and title and abstract screening, as per the study's inclusion criteria. All articles that potentially met eligibility criteria were retrieved after a full-text screen of all potential articles. After consulting with the other three authors, all articles meeting the eligibility criteria were retained.

### Search strategies

The search was conducted in English and performed in PubMed, CINAHL Complete, APA PsycINFO, Web of Science, Scopus, and Google Scholar in August 2023, covering articles published in or before 2023. The search strategy included Medical Subject Headings (MeSH) terms, keywords, and free terms related to the systematic review. The Boolean operators–OR and AND–were instrumental with the search terms in identifying the included studies. The

**Table 1. List of keywords for identifying articles for adverse childhood experience–elder abuse victimization systematic review.**

| Category 1- Adverse childhood experience (ACEs) | Category 2—Elder abuse victimization | Category 3 –Older adults |
|---|---|---|
| **#1 Childhood factors** | Elder abuse | Older people |
| Childhood adversities | ageism | Elderly |
| Child ageism | physical assault* | Elders |
| Childcare abuse | psychological aggression* | Seniors |
| Childhood neglect | active neglect* | Senior citizens |
| Childhood sexual abuse | passive neglect* | Aged |
| Childhood emotional abuse | Sexual abuse* | Frail |
| Childhood abuse | Active neglect* | Older persons |
| Childhood neglect | Abandonment* | |
| Childhood trauma | financial abuse* | |
| childhood maltreatment | | |
| Childhood vulnerability | | |
| Childhood disempowerment | | |
| Childhood discrimination | | |
| **#2 Parental/family factors** | | |
| Parental criminality | | |
| Parental divorce | | |
| Parental mental illness | | |
| Parental violence | | |
| Economic adversity | | |
| Family violence | | |
| Parent Past abuse | | |
| Parent past trauma | | |

The initial search strategy was developed in PubMed and later searched in the other five databases including CINAHL Complete, APA PsycINFO, Web of Sciences, Scopus, and Google Scholar. The keywords with * were searched with category 3 keywords with OR and the results were combined with category 1 keywords.

search strategy used the following Boolean descriptors with all keywords under each of the categories described in Table 1. Like keywords were searched separately with "OR" for each category (1 and 2) and "AND" was used to combine all categories. The keywords with * were searched with category 3 keywords with OR and the results were combined with category 1 keywords.

## Study selection

After the database search, the total number of studies (N = 9,453) was exported to Endnote for the removal of duplicates. After this process, two authors (KAN and KAC) independently did title and abstract screening of the remaining articles (N = 8,653) to determine their eligibility. Where there was disagreement between these authors, the second and fifth authors (BS and DD) resolved them via consultation. Eventually, 9 studies met the eligibility criteria and were included in synthesis and meta-analysis where applicable (See Fig 1).

## Quality assessment

The Joanna Briggs Institute appraisal checklists for analytical cross-sectional and cohort studies were used to assess the methodological quality of the included papers [24, 25]. The checklists were made up of 8 and 11 questions for analytical cross-sectional and cohort studies, respectively. The results of the methodological quality assessment are presented in Table 2.

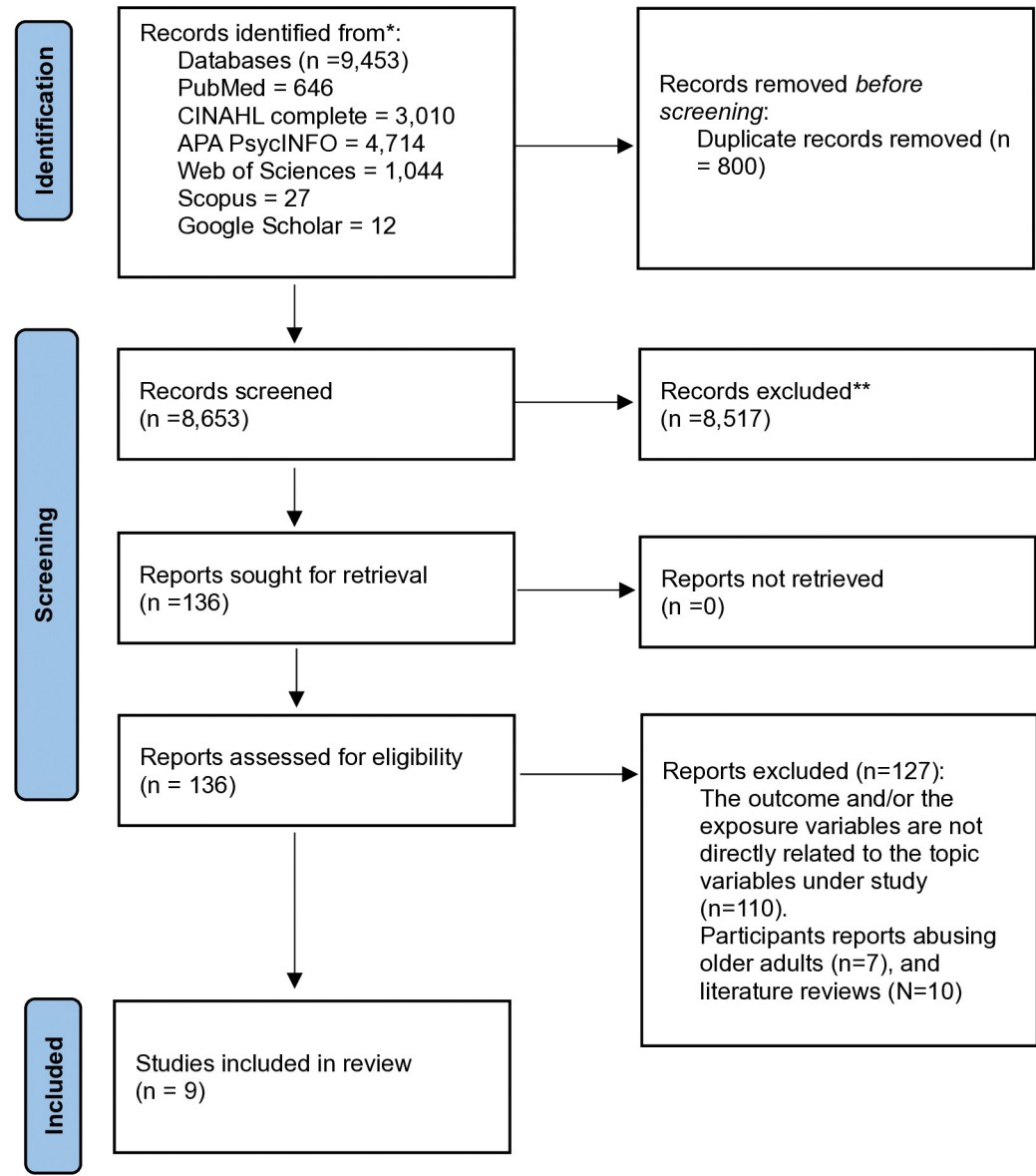

**Fig 1. Study inclusion procedure in line with the PRISMA 2020 guidelines.**

## Data extraction and analysis

After eligible studies were identified, two authors (KAC and KAN) independently did the initial data extraction using an extraction tool developed by the authors. The third author (NA) reviewed all extracted data and cross-checked the original studies for a thorough and comprehensive data extraction. Any disagreement between the two reviewers (KAC and KAN) was resolved via a consultation with the second (BS), the third (NA), and the fifth author (DD).

Examples of study details extracted were (authors, year, sample size, country, and study design as well as participants' details such as age range, and sex), the outcome, and control variables measured in the studies (See Table 3).

A narrative synthesis and meta-analysis were used to analyze the findings. Using the odds ratios (ORs) and confidence intervals (CIs) of the included studies, findings were interpreted

**Table 2. Results of quality assessment of the selected articles.**

| Study | Checklist questions | | | | | | | | | | | Total | % |
|---|---|---|---|---|---|---|---|---|---|---|---|---|---|
| | Q1 | Q2 | Q3 | Q4 | Q5 | Q6 | Q7 | Q8 | Q9 | Q10 | Q11 | | |
| Asyraf et al. | 1 | 1 | 1 | 1 | 1 | 1 | 1 | 1 | | | | 8 | 100% |
| Burnes et al. | 1 | 1 | 1 | NA | NA | NC | 1 | NC | NC | NC | 1 | 5 | 45% |
| Chen & Fu | 0 | 1 | 1 | 1 | 0 | 0 | 1 | 1 | | | | 5 | 63% |
| Dong & Wang | 1 | 1 | 1 | 1 | 1 | 1 | 1 | 1 | | | | 8 | 100% |
| Easton et al. | 1 | NC | 1 | 1 | 1 | NC | 1 | 1 | NC | NC | 1 | 7 | 64% |
| Giraldo-Rodríguez et al. | 1 | 1 | 1 | 0 | 1 | 1 | 1 | 1 | | | | 7 | 88% |
| Kong & Easton | 1 | 1 | 1 | 1 | 1 | 1 | 1 | 1 | | | | 8 | 100% |
| McDonald & Thomas | 1 | 1 | 1 | 1 | 1 | 1 | 1 | 1 | | | | 8 | 100% |
| Wiklund et al. | 1 | 1 | 1 | 1 | 1 | 1 | 1 | 1 | | | | 8 | 100% |

**Note**: the results in the table are based on the JBI (Joanna Briggs Institute) checklists for cross-sectional and cohort studies; 1 represents "yes" (i.e., the checklist item was met by the paper), 0 represents "no" (i.e., checklist item was not met by the paper); NC–not clear (i.e., there was insufficient information in the paper to decide), NA–not applicable (i.e., the checklist item is not applicable due to the nature of the study); studies corresponding to eight questions are cross-sectional studies whereas studies corresponding to 11 questions are cohort studies; % represents the proportion of checklist items met for each paper; higher % score indicates higher quality of the paper.

as having a statistically significant association between outcome measures–EAVs measures and control variables–ACEs measures. A narrative was employed to group similar findings based on the number of studies reporting an association with the specific association nexus. This grouping generated themes providing structure for the presentation of the findings.

After the narrative review was completed for a particular theme, a meta-analysis of the extracted ORs and CIs was conducted using the common-effect inverse-variance model in STATA 16.1. Pooled ORs, CIs, and heterogeneity tests were used to present consolidated associations for the themes identified, where applicable. When multiple measures were reported in the same study, we included all in the meta-analysis. Using Stata 16.1, we employed $I^2$ to assess the presence of heterogeneity in results across studies, and when $I^2$ is greater than 40%, a random-effects model was used due to the presence of significant heterogeneity [26].

## Results

The literature review search resulted in 9,453 articles, with a total of 800 duplicates removed which resulted in 8,653 for screening (see S2 File). After title and abstract screening were completed, a total of 136 articles were assessed for eligibility via a full-text screening (see Fig 1). The full-text screening resulted in 9 articles (all used quantitative designs) that met eligibility criteria for the systematic review and meta-analysis where appropriate.

### Included study characteristics

Table 3 details the characteristics of the included studies. These studies reported an association between ACEs and EAV. Studies were mostly conducted in the Global North (N = 7) [9, 10, 14, 18–20, 27, 28], predominantly in the United States of America (N = 3) [9, 27, 28] followed by Canada (N = 2) [18, 19]. The rest of the studies were conducted in the Global South particularly China and Malaysia (N = 2) [14, 29]. None were conducted in an African country. Of the nine studies, three were conducted in an institutional setting, such as health and education, whereas the remaining (N = 6) were conducted in the community.

The sample sizes of the included studies ranged from 135 to 23,468, and participants were predominantly females.

**Table 3. Details of studies included in the systematic review.**

| Study | Country | Sample size | % Female | Age, mean | Study design | Adverse childhood experience measure | Elder abuse victimization measure | Results |
|---|---|---|---|---|---|---|---|---|
| Asyraf, Dunne [14] | Malaysia | 1,984 | 54.7% | ≥ 60 | Cross-Sectional study | Adverse Childhood Experience Questionnaire (ACE-IQ) designed by the World Health Organisation Examples of the 13 items include emotional abuse. physical abuse; sexual abuse; violence against household members; living with household. members who were substance abusers; living with household members who were mentally ill or suicidal; living with household members who were imprisoned; growing up with one or no parents, parental separation or divorce during childhood; emotional neglect; physical neglect; Bullying; exposure to community violence; and exposure to collective violence. | Elder abuse victimisation using 27 items in the National Elder Abuse and neglect study from Ireland adapted from Revised Conflict Tactics scales–Cumulative Single measures including: Physical elder abuse Psychological elder abuse Financial elder abuse Sexual elder abuse | Reporting three ACEs (OR = 2.32, 95% CI 1.56, 3.47) increased risk of elder abuse victimisation. Reporting four or more ACEs (OR 1.71, 95% CI 1.14, 2.56) increased the risk of elder abuse (overall abuse experience). Reporting any two ACEs had no association with elder abuse victimisation in later life (OR = 1.25, 95%CI 0.90, 1.74) Reporting any one ACE had no association with elder abuse victimisation in later life (OR = 1.25, 95%CI 0.90, 1.74) Reporting four or more ACEs was associated with any abuse type experienced in later life (OR = 1.70, 95% CI, 1.6, 2.47). Reporting three ACEs was associated with any abuse type experienced in later life (OR = 2.67, 95%CI, 1.84, 3.87). Reporting two ACEs was associated with financial abuse in later life (OR = 1.40, 95%CI, 1.01, 1.94) Reporting three ACEs was associated with financial abuse in later life (OR = 2.54, 95%CI, 1.73, 3.75) Reporting three ACEs was associated with psychological abuse in later life (OR = 2.30, 95%CI, 1.28, 4.16) Reporting four or more ACEs was associated with psychological abuse in later life (OR = 2.16, 95%CI = 1.21, 3.86) Reporting four or more ACEs was associated with physical abuse in later life (OR = 14.1, 95%CI, 1.57, 127.1) Reporting four or more ACEs was associated with sexual abuse in later life (OR = 8.94, 95%CI 1.05, 75.8) |
| Burnes, Pillemer [19] | Canada | 23,468 | 52.6% | 69.4 | Cohort Study | Child maltreatment using 9 mistreatment behaviours or items measured from a modified version of the Childhood Experiences of Violence Questionnaire | Cumulative Elder Abuse measured from three measures or subtypes including. Emotional elder abuse (using four items) Physical elder abuse (using six items Financial elder abuse (using three items) | A rise in each child maltreatment level (36 levels) had an association with increased in elder Abuse (multiple) (OR, 1.01; 95% CI, 1.01–1.01). A rise in each child maltreatment level in childhood increased the risk of emotional abuse in later life (OR, 1.01; 95% CI, 1.01–1.01), A rise in each child maltreatment level in childhood increased the risk of physical abuse in later life (OR, 1.01; 95% CI, 1.01–1.02) A rise in each child maltreatment level in childhood was associated with increased risk of financial abuse in later life (OR, 1.01; 95% CI, 1.01–1.02). Each increase in ACE (3 total experiences) had a significant association with financial abuse in later life (OR, 1.30; 95% CI, 1.08–1.58). |

**Table 3.** (Continued)

| Study | Country | Sample size | % Female | Age, mean | Study design | Adverse childhood experience measure | Elder abuse victimization measure | Results |
|-------|---------|-------------|----------|-----------|--------------|--------------------------------------|-----------------------------------|---------|
| Chen and Fu [29] | China | 1,002 | 53.6% | 74.9 | Cross-Sectional study | Retrospective self-report items (15 questions) were used to measure Adverse Childhood experience of participants prior to turning 18 years old (some of the items include socio-economic difficulty of original family, parental divorce, frequent quarrels between parents, frequent physical punishment by parents, starvation etc | Elder abuse victimisation (6-item measure) Participants' victimization experiences during old age were measured using six self-constructed items, including experiencing physical assault, psychological. Aggression, active neglect, passive neglect, abandonment, and financial abuse. The physical and psychological aggression items were developed based on the revised Conflict Tactics Scales (CTS2) | Having at least 4 ACEs was a risk factor for elder abuse victimization (OR = 3.06, 95%CI, 1.77, 5.30). Frequent physical punishment by parents was associated with a higher risk of elder abuse victimisation (OR = 2.29, 95%CI, 1.12, 4.70). Poor socio- economic status of the original family was associated with a higher risk of elder abuse victimisation (OR = 1.76, 95%CI, 1.02, 3.04) Starvation in childhood had no significant association with elder abuse in later life (OR = 1.58, 95%CI 0.83, 2.99) Parental divorce during childhood had no significant association with elder abuse in later life (OR = 3.02, 95%CI 0.83, 10.9). Frequent quarrels between parents during childhood had no significant association with elder abuse in later life (OR = 1.98, 95%CI 0.95, 4.09) |
| Dong and Wang [27] | USA | 3,157 | 56.4% | $\geq$ 60 | Cross-Sectional study | Child maltreatment before the age of 18years (measure with 5-items Extended Hurt, Insult, Threaten, Scream scale | Elder abuse experience (10-items) a self-reported measure modified from the Hwalek-Sengstok Elder Abuse Screening Test and the Vulnerability to Abuse Screening Scale | Reporting childhood maltreatment experience had increased odds of elder abuse victimisation (OR = 2.08; 95% CI, 1.57–2.75) |
| Easton and Kong [9] | USA | 5,968 | 53.47% | 71 | Cohort study | Childhood Adversities (7-item measure—emotional abuse, physical abuse, neglect, family dysfunction, parental divorce, witnessing domestic violence, and living with a household member with a substance problem) | Elder abuse victimisation measured using Abusive Behaviour Inventory | Childhood adversities had significant association with elder abuse victimization (0.11*, p<0.001). Childhood adversities also had an indirect implication on elder abuse through physical health (0.00*, p<0.001) and depressive symptoms (0.01*, p<0.001 |
| Kong and Easton [28] | USA | 5,968 | 53.5% | 71 | Cross-Sectional study | Childhood neglect (1 item) Childhood physical/verbal abuse (2 items) Childhood sexual abuse (4 items) | Elder abuse victimization (using Abusive Behavior Inventory | Frequent exposure to childhood emotional abuse increased the risk of elder abuse victimization (OR = 1.37, 95% CI, 1.06, 1.77) The odds of elder abuse victimization were about two times higher for persons with a history of child sexual abuse compared with those with no such reported history (OR = 3.34, 95% CI, 1.88, 5.94). Childhood neglect had no significant association with elder abuse victimization (OR = 1.00, 95%CI, 0.92, 1.10) Childhood physical abuse was not significantly associated with elder abuse victimization (OR = 1.18, 95%CI 0.96, 1.44) |
| McDonald and Thomas [18] | Canada | 267 | 76.8% | $\leq$ 75 | Cross-Sectional study | Childhood psychological abuse Childhood physical abuse Childhood sexual abuse | Psychological abuse, Physical abuse, Sexual abuse, financial abuse | History of childhood abuse increased the risk of experiencing one type of abuse in later life (OR = 1.81, 95% CI, 1.01–3.26). |

*(Continued)*

**Table 3.** (Continued)

| Study | Country | Sample size | % Female | Age, mean | Study design | Adverse childhood experience measure | Elder abuse victimization measure | Results |
|---|---|---|---|---|---|---|---|---|
| Giraldo-Rodríguez, Mino-León [10] | Mexico | 18,416 | 100% | ≥ 60 | Cross-Sectional study | Childhood psychological, Childhood physical, Childhood sexual abuse (occurred before age 15) | Four categorical items (psychological, economic, physical, and sexual abuse) during the last 12 months by a relative or acquaintance. | Reporting a combination of psychological and sexual childhood abuse experience increased the odds of experiencing elder abuse victimisation (OR = 3.89; 95% CI, 3.09, 4.91). Reporting a combination of psychological, physical, and sexual childhood abuse increased the odds of experiencing elder abuse victimisation (OR = 3.55, 95% CI; 3.16,3.99) Reporting a combination of physical and sexual childhood abuse increased the odds of experiencing elder abuse victimisation (OR = 2.62; 95% CI, 2.40, 2.88). Reporting a combination of psychological and physical childhood abuse increased the odds of experiencing elder abuse victimisation (OR = 2.21, 95% CI 2.0, 2.37) Reporting a childhood psychological abuse increased the odds of experiencing elder abuse victimisation (OR = 2.21, 95% CI, 1.94, 2.50) Reporting physical childhood abuse increased the odds of experiencing elder abuse victimisation (OR = 1.28, 95% CI, 1.17, 1.42) Reporting childhood sexual abuse increased the odds of experiencing elder abuse victimisation (OR = 2.32, 95% CI, 2.02, 2.64) Those who experienced one type of childhood abuse were 1.46 more likely to experience elder abuse victimisation in later life (OR = 1.46, 95% CI, 1.35, 1.58) Who experienced two types of childhood abuse were 2.27 times more likely to experience revictimization in later life (OR = 2.27, 95% CI, 2.10, 2.47) Those who experienced three types of childhood abuse were 3.58 times more likely to experience revictimization in later life (OR = 3.58, 95% CI, 3.04, 4.23) Women who reported having experienced three types of child abuse increased their risk of elder abuse. |
| Wiklund, Ludvigsson [20] | Sweden | 135 | 54.1% | >65 | Cross-Sectional study | Abuse before the age of 65years | Elder abuse victimization | Being abused before the age of 65 as a background factor was associated with an increased risk of elder abuse in later life (OR = 5.4; 95% CI 1.9–15.7). |

## ACEs measures

Included studies revealed varying forms of ACE measures. Some studies measured ACEs cumulatively [9, 14, 27, 29] while others measured forms of ACEs separately [10, 18–20, 28]. The studies reported using different types of measurement tools with unequal and unique sets of questions to measure ACEs. The studies included the following questionnaires: the ACEs

International Questionnaire (ACE-IQ) comprised of 29 items [14]; the Child Maltreatment Questionnaire that measured 36 items [19]; a retrospective self-report (15 questions) [29]; a 5-item child maltreatment questionnaire [27]; the childhood adversities questionnaire (7-item measure) [9]; a childhood neglect (1-item), physical/verbal abuse (2-item), and sexual abuse (4-item) questionnaire [28]; the childhood psychological, physical, and childhood sexual abuse questionnaire [18]; the childhood psychological, physical, and sexual abuse before age 15 questionnaire [10] and a questionnaire measuring abuse before the age of 65 [20].

## EAVs measures

Even though multiple EAVs was the most common outcome measure among the studies [9, 14, 19, 28, 29], single forms of EAV were reported, as well, including physical elder abuse [10, 14, 18, 19], financial elder abuse [14, 18, 19], psychological elder abuse [10, 14, 18], sexual elder abuse [10, 18, 19], emotional elder abuse [10, 19], and economic elder abuse [10].

The included studies used distinct scales and question items to measure EAV, including the Abusive Behavior Inventory [9, 28]; a self-reported measure modified from the Hwalek-Sengstok Elder Abuse Screening Test and the Vulnerability to Abuse Screening Scale (10 items) [27], six self-constructed items [29]; and a three items (emotional, physical, and financial elder abuse) questionnaire [19]. The 27 items in the National Elder Abuse and Neglect Study from Ireland were adapted from the Revised Conflict Tactics Scales [14], and four self-reported items [10, 18]. One of the included studies did not indicate the EAV measure it used [20].

## Connections between single or multiple ACEs and multiple EAV

Seven studies examined the association between multiple ACEs and multiple EAVs. The four studies that examined the association between multiple ACEs and multiple EAV showed a consensus direction of the relationship. Burnes, Pillemer [19] and Dong and Wang [27] reported that a history, or increasing incidence of child maltreatment, was associated with an increase in EAV in later life. Similarly, experiencing multiple ACEs had a significant positive association with multiple EAVs [9]. Moreover, a history of abuse of any kind before the age of 65 years was associated with a high chance of EAV [20]. The pooled odds ratio from a meta-analysis shows a statistically significant association between experiencing ACEs and EAV (OR = 2.21, 95%CI: 1.69, 2.90), with a slight heterogeneity existing across studies ($I^2$ = 65.9%; $p$ = 0.087) (see Fig 2).

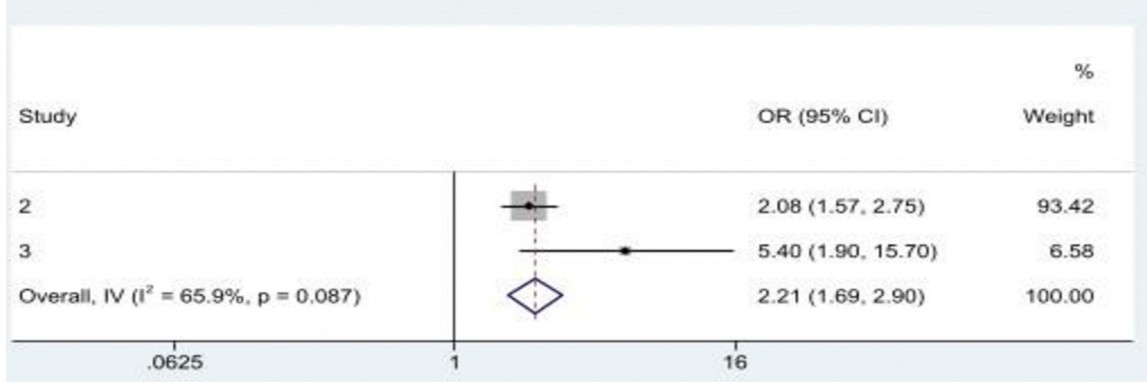

**Fig 2. Multiple ACEs was associated with multiple EAVs in later life. Note:** *Study 2 = Dong and Wang (2019); Study 3 = Wiklund et al. (2022).*

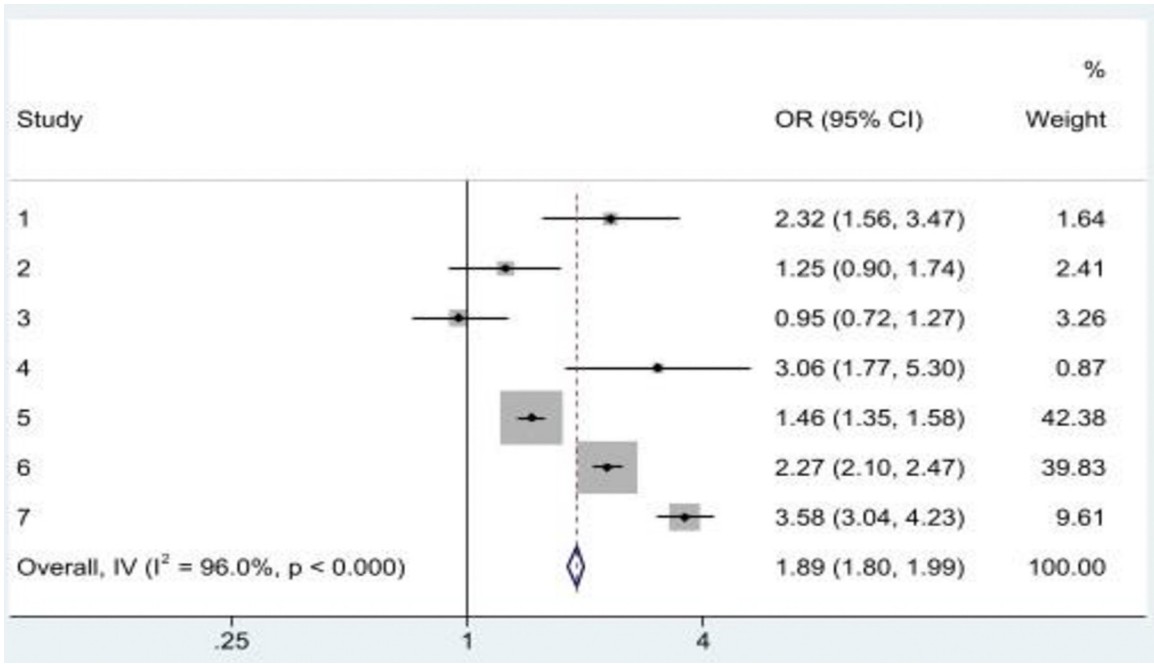

**Fig 3. A history of at least one ACE was significantly associated with multiple EAVs. Note:** Study 1, 2 & 3 = Asyraf et al. (2021); Study 4 = Chen and Fu (2022); Study 4, 6 & 7 = Giraldo-Rodríguez et al. (2022).

Some studies examined the effect of having a history of co-occurring ACEs on EAV in later life. While most of the results were consistent, some contradictions were apparent. Experiencing either three or at least four ACEs was associated with EAV [10, 14, 29]. While a significant association with EAV was found when an older adult experienced one or two ACEs [10], Asyraf, Dunne [14] found no significant association. Specifically, older adults who had experienced psychological-sexual, psychological-physical-sexual, physical-sexual, and psychological-physical childhood abuse combinations experienced a heightened risk of EAV in later life [10]. Pooling odds ratios in a meta-analysis (Fig 3), there was a statistically significant positive association between a history of at least one ACE and multiple EAVs (OR = 1.89, 95%CI: 1.80, 1.99). Despite this, heterogeneity existed across studies ($I^2$ = 96.0%; $p$<0.001) signifying a great variability in the results.

Three studies examined the association between reporting a single form of ACE and multiple EAVs in later life. In the US and Mexico, reports of the association between a history of childhood physical abuse and an increased risk of multiple EAVs in later life differed [10, 28]. While in the Mexican study, childhood physical abuse among women was associated with multiple EAVs in later life [10], a lack of statistical significance was reported in the US [28]. In both Mexico and the US, childhood sexual abuse was significantly associated with multiple EAVs in later life [10, 28]. Childhood neglect, frequent exposure to childhood emotional abuse [28], childhood psychological abuse [10], frequent physical punishment by parents, and poor socio-economic status of the original family [29] singly increased the risk of multiple EAVs in later life. Parental divorce during childhood did not have a significant association with multiple EAV in later life [29]. Fig 4 shows the pooled OR of the association between a history of single form of ACE and multiple EAVs in later life (OR = 1.42, 95%CI: 1.35, 1.49; $I^2$ = 93.8%; $p$<0.001).

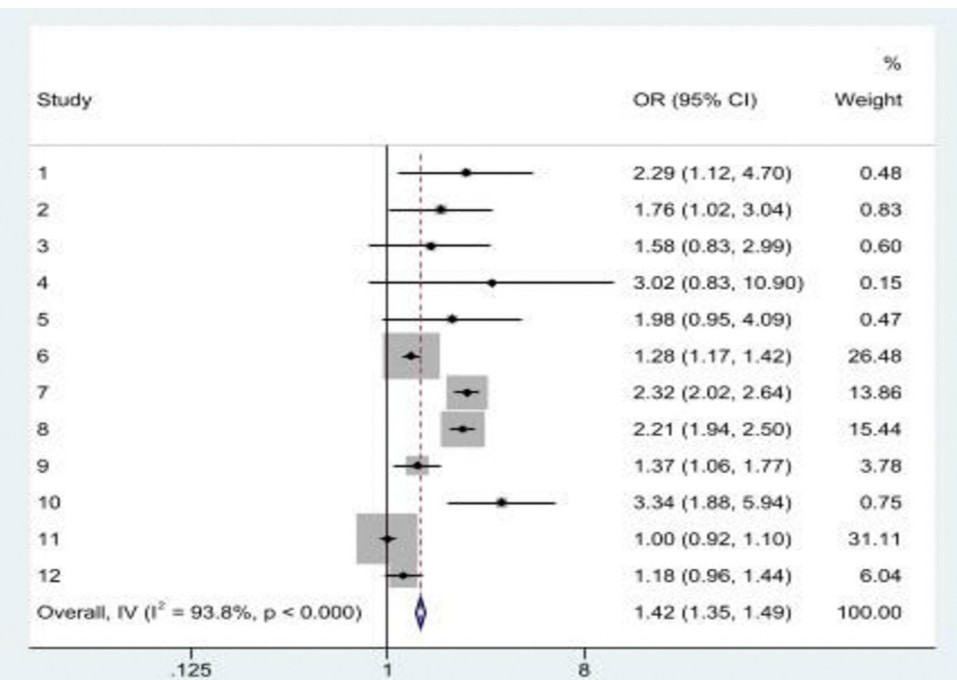

**Fig 4. A history of single forms of ACE was significantly associated with multiple EAVs in later life. Note:** Study 1, 2, 3, 4 & 5 = Chen and Fu (2022)); Study 6, 7 & 8 = Giraldo-Rodríguez et al. (2022); Study 9, 10, 11 &12 = Kong and Easton (2019).

## Connections between multiple ACEs and a single form of EAV

Overall, two studies examined the association between multiple ACEs and a single form of EAV and both reported that multiple ACEs were associated with the risk of experiencing a single form of EAV in later life [14, 18]. In addition, Asyraf, Dunne [14] reported that experiencing at least three ACEs was associated with a single form of EAV in later life. The pooled OR in a meta-analysis (Fig 5) showed a statistically significant positive association between multiple

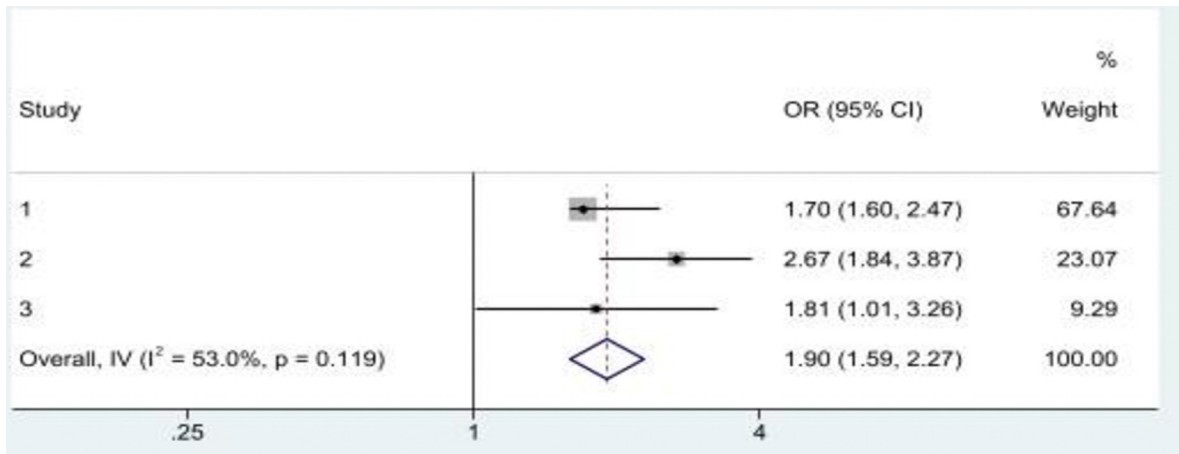

**Fig 5. A history of multiple ACEs was significantly associated with a single form of EAV. Note:** Study 1 & 2 = Asyraf et al. (2021); Study 3 = McDonald & Thomas (2013).

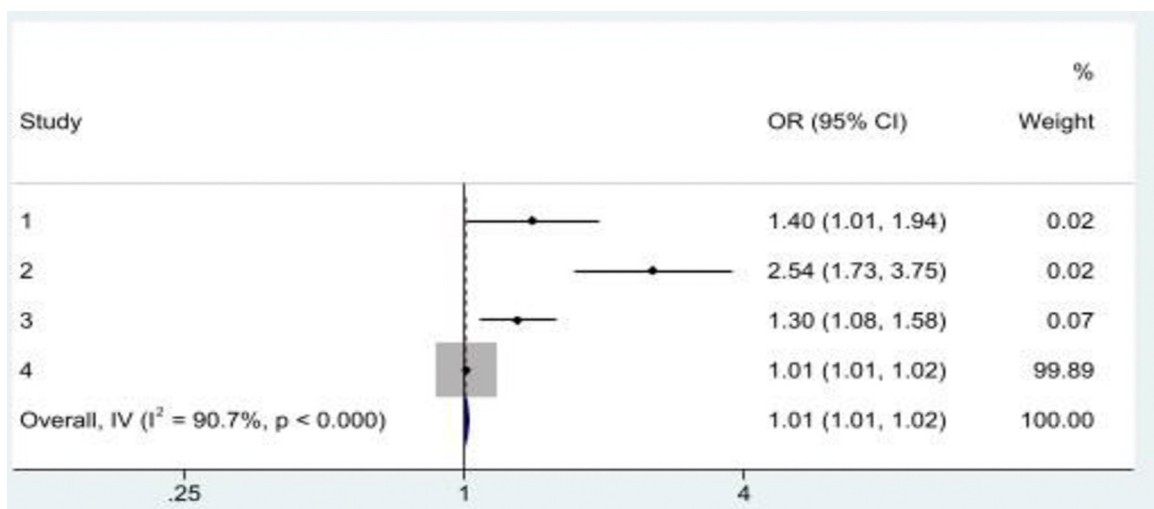

**Fig 6. A history of multiple ACEs was associated with financial elder abuse in later life. Note:** Study 1 & 2 = Asyraf et al. (2021); Study 3 = Burnes et al. (2022)); Study 4 = Chen & Fu (2022).

ACEs and a single form of EAV (OR = 1.89, 95%CI: 1.80, 1.99), with slight heterogeneity across studies ($I^2$ = 53.0%; $p<$0.119).

## Connections between multiple ACEs and financial abuse experience in later life

Three studies examined the association between multiple ACEs and financial elder abuse experience in later life [14, 19, 29]. All three studies showed a consensus result that an increase in ACEs or the occurrence of an ACE was associated with financial elder abuse in later life [14, 19, 29]. Specifically, a study conducted in Canada reported that an increase in ACE levels had a significant positive association with financial elder abuse experiences in later life [19]. Similarly, a Malaysian study further found that experiencing two or three ACEs was associated with financial elder abuse in later life [14]. In China, a study that measured the association between child maltreatment–a form of ACEs–reported that an increase in child maltreatment levels was associated with an increased risk of financial elder abuse experience in later life [29]. When the ORs were pooled in a meta-analysis (Fig 6), there was a statistically significant positive association between multiple ACEs and financial elder abuse in later life (OR = 1.01, 95% CI: 1.01, 1.02), with significant heterogeneity across studies ($I^2$ = 90.7; $p<$0.001).

## Connections between multiple ACEs and physical abuse experience in later life

Two studies [14, 29] specifically examined the association between ACEs and physical elder abuse, and both studies revealed a significant positive association between ACEs and physical elder abuse. The Chen and Fu [29]'s study conducted in China reported that an increase in the incidence of child maltreatment in childhood increased the risk of physical elder abuse in later life. The Asyraf, Dunne [14]'s study in Malaysia reported that an association of ACEs with EAV depended on the number of ACEs experienced; a childhood experience of four or more ACEs was associated with physical elder abuse [29]. The pooled ORs in a meta-analysis (Fig 7) showed a statistically significant positive association between multiple ACEs and physical elder abuse in later life (OR = 1.01, 95%CI: 1.01, 1.02), with significant heterogeneity across studies ($I^2$ = 81.9%; $p<$0.05).

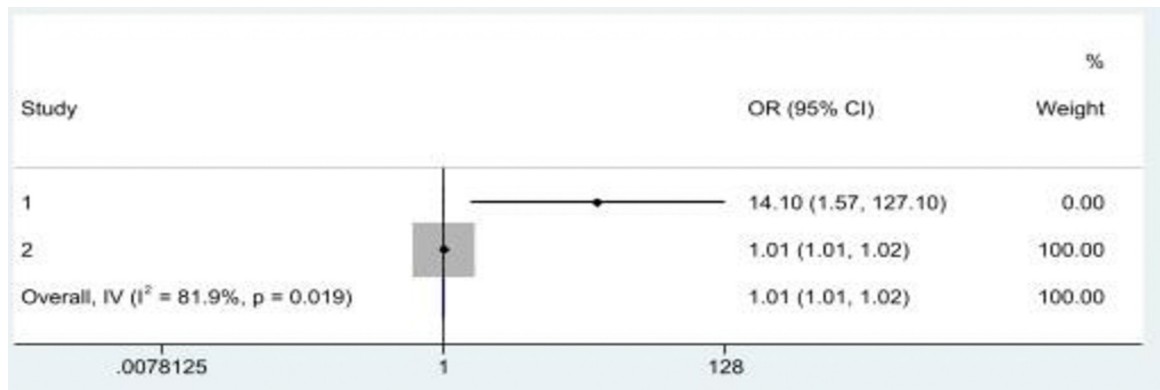

**Fig 7. A history of multiple ACEs was significantly associated with physical elder abuse in later life.** Note: Study 1 = Asyraf et al. (2021) & Study 2 = Chen and Fu (2022).

### Connections between multiple ACEs and emotional, psychological, and sexual elder abuse experiences in later life

Two studies explored the association between multiple ACEs and emotional, psychological, and sexual elder abuse altogether [14, 29]. One of these studies reported that an increased incidence of childhood maltreatment elevated the risk of emotional elder abuse experienced in later life [29]. Asyraf, Dunne [14] reported that three or more ACEs was associated with psychological elder abuse in later life. Moreover, reporting four or more ACEs was associated with sexual elder abuse in later life [14].

### Moderating factors of the connections between ACE and EAV

Two studies explored the factors that moderated the association between ACE and EAV [9, 10]. Giraldo-Rodríguez, Mino-León [10] found that women who reported having experienced three types of child abuse were at heightened risk of EAV. In addition, poor physical health and depressive symptoms moderated the association between ACEs and EAV [9].

## Discussion

In this study, we examined the existing association between ACEs and EAV and employed a life course perspective to explore the relevance of addressing childhood and middle adulthood factors that contribute to EAV. We sourced the data from nine published articles that met the inclusion criteria for the study. We consider the nine selected papers as low volume, indicating limited scholarly conversation on the association between ACEs and EAV. This limited conversation has ongoing implications for policies and programs addressing elder abuse and child protection. This low volume of papers contributes to understanding why practices and policies tackling elder abuse have lagged despite its effects [8, 30].

The results further show different measures of ACEs and EAV, raising questions about the reliability and validity of measures. For instance, only one study used a standardized instrument (International questionnaire (ACE-IQ)) to measure ACEs [14], whereas the remaining studies used different instruments to measure an aspect of ACEs such as childhood neglect, childhood physical, and childhood sexual abuse [10, 28]. Likewise, different measures such as the Revised Conflict Tactics Scales [14], and Abusive Behaviour Inventory [9] composed of different questions and domains were used to measure EAV. Due to these inconsistencies, comparisons of study findings across time and a variety of different settings might be difficult to achieve, warranting the need for a standardized measurement approach that has relevance to a broader range of studies on ACEs.

Furthermore, ACEs were associated with the risk of experiencing a single form of EAV in later life. In particular, the selected studies established a significant positive association between ACEs and financial elder abuse experience, physical elder abuse, and emotional, physical, and sexual elder abuse. Our results further show that gender, physical health, and depressive symptoms moderate the association between ACEs and EAV. Specifically, an older woman with an experience of multiple ACEs or any one form of ACE is more likely to experience EAV in later life. Likewise, living with poor physical health with an experience of single or multiple ACEs increases the likelihood of experiencing EAV in later life. Moreover, expressing depressive symptoms and having a previous single or multiple ACEs increase the risk of experiencing EAV. Even though the present study has shown some evidence that gender, physical health status, and presence of depressive symptoms moderate ACEs–EAV association, only two studies were identified that support these findings. These moderated associations should, therefore, be considered by researchers, policymakers, and practitioners with care. The paucity of evidence to support the moderating powers of gender, physical health status, and depression on the ACEs–EAV association calls for significantly greater research attention from different geographical settings. Irrespective of the fact that this evidence has not been firmly established, interventions tackling EAV should attend to the unique needs of women/girls, those living with poor physical health, and those with depression.

The evidence from our analysis is consistent with other theoretical and empirical literature on elder abuse. Theoretically, the association between ACEs and EAV aligns with the life-course perspective, which argues that events occurring at different stages of life, especially ACEs, are risk factors for elder abuse [18, 30]. For example, early life events, including childhood abuse, childhood exposure to domestic violence, and childhood neglect heighten the risk of EAV in later life [28, 30]. Storey [31] describes ACEs as prior victimization and constant factors that increase the risk of elder abuse victimization.

From a life course perspective, the association between ACEs and EAV reinforces the need for intervention in childhood to prevent the occurrence of ACEs and mitigate adversities stemming from ACEs throughout life. Interventions addressing ACEs over the life stages, such as exposure to family violence, will likely impact the incidence of EAV [32]. However, there is no evidence of a prospective longitudinal study that tracks specific child abuse prevention interventions and their effect on the incidence of EAV in later life. Rather current interventions are mostly remedial for children whose effects are uncertain and inconclusive [33].

The systematic review and meta-analysis have strengths and limitations. A key strength is the consolidation of research on the strength and nature of association between ACEs and EAV. The current study can motivate and inform the design of more longitudinal studies to examine the associations between ACEs and EAV, in single or multiple forms, as well as their moderating factors in these relationships. A key limitation is the small number of studies eligible for inclusion in this study. Though nine studies collectively comprise a total sample of 60,365, a larger number of studies could have produced more robust findings. Moreover, though there was no study excluded based on grey literature status, an exclusion criterion might have automatically excluded grey literature relevant to this analysis. Additionally, the scope of the conversation is limited to the number of available published research evidence and the global North, calling for more longitudinal studies on ACEs–EAV associations and moderating factors in different geographical contexts including developing countries.

## Conclusion and implications

There is growing evidence of an association between ACEs and EAV moderated by gender, and physical and mental health. Given the evidence of the association between ACEs and EAV

demonstrated in our analysis, it is important to develop interventions that prevent ACEs early in life and that support older adults living in both residential and independent homes. Reducing the incidence of ACEs in childhood will likely also reduce the incidence of EAV or re-victimization as supported by the life course perspective. A life course perspective propels the need for effective support and services for children and families, as the prevention of adversity in childhood will also positively impact elder abuse prevention.

## Critical findings

- There is a growing evidence of an association between ACEs and EAV moderated by gender, physical and mental health

- A low volume of research on the association between ACEs and EAV limits scholarly discussion on this topic.

- Consistency in the measurement of ACEs and EAV is lacking, raising questions about the reliability and validity of measures.

- There is no evidence of a prospective longitudinal study that tracks specific child abuse prevention interventions and their effect on the incidence of EAV

## Implications for practice, research, and policy

- Practitioners who work with children and older adults should be conscious of developing interventions that prevent ACEs early in life and that support older adults who live in either residential or independent homes.

- The current study has provided foundational evidence of the need for more longitudinal studies to examine the associations between ACEs and EAV, in single or multiple forms, as well as their moderating factors to understand how to better serve the changing needs of older adults and children.

- Policymakers should apply a life course perspective, which propels the need for effective support and services for children and families who experience victimization and its consequences at all stages of life.

## Supporting information

**S1 File. PRISMA 2020 checklist: The PRISMA checklist for systematic reviews and meta-analyses.**
(DOCX)

**S2 File. Studies identified.**
(XLSX)

## Author Contributions

**Conceptualization:** Kofi Awuviry-Newton, Bernadette Saunders.

**Data curation:** Kofi Awuviry-Newton, Kwamina Abekah-Carter, Daniel Doh.

**Formal analysis:** Kofi Awuviry-Newton, Nestor Asiamah.

**Investigation:** Kofi Awuviry-Newton.

**Methodology:** Kofi Awuviry-Newton, Bernadette Saunders, Nestor Asiamah, Kwamina Abekah-Carter, Daniel Doh.

**Project administration:** Kofi Awuviry-Newton.

**Resources:** Kofi Awuviry-Newton.

**Software:** Kofi Awuviry-Newton.

**Supervision:** Kofi Awuviry-Newton.

**Validation:** Kofi Awuviry-Newton, Bernadette Saunders.

**Visualization:** Kofi Awuviry-Newton, Bernadette Saunders.

**Writing – original draft:** Kofi Awuviry-Newton, Daniel Doh.

**Writing – review & editing:** Kofi Awuviry-Newton, Bernadette Saunders, Nestor Asiamah, Kwamina Abekah-Carter, Daniel Doh.

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
