## [Decision Letter · Decision Letter 0]

25 Jul 2024

PONE-D-24-07611

Adverse childhood experiences and elder abuse victimization nexus: A systematic review and meta-analysis

PLOS ONE

Dear Dr. Awuviry-Newton,

Thank you for submitting your manuscript to PLOS ONE. After careful consideration, we feel that it has merit but does not fully meet PLOS ONE’s publication criteria as it currently stands. Therefore, we invite you to submit a revised version of the manuscript that addresses the points raised during the review process.

Please address Reviewer 1 numbered comments as follows:

1. Provide a more concise overview of the findings in the abstract.

2. Provide a citation supporting the claim of a higher prevalence of EAV in low- and middle-income countries compared to higher income countries.

3. The sentence in question reads as if there is only one pathway from ACEs to EAV, which is not true. Please revise this sentence to reflect that the pathway described is one of several potential pathways, of particular relevance to the present study, and provide relevant citations. 

4. I believe this paragraph is relevant. However, please shorten and focus on other systematic reviews linking ACEs to other outcomes and to the highlighted claim regarding the absence of systematic reviews on ACEs and EAV.

5. Please be sure to temper the claim that there has been "a significant number of bodies of research on this association". The outcome of the systematic review itself suggests that this is not the case and contradicts a later claim that there is a "low volume" of scholarship on this topic in research (Critical finding, page 20).

6. Please thoroughly review the submission for typos.

7. The criteria are met for a systematic review.

8. Please expand discussion on measurement inconsistencies and use more caution in the discussion about moderating factors, as described by the Reviewer.

9. Please expand the discussion on the justification for applying a life course perspective, as described by the Reviewer.

10. Be sure to address the comments of Reviewer 1 in terms of what to emphasize in more detail. I would also like to see a brief mention of possible limitations of excluding grey literature from the search process. 

Please address Reviewer 2 comments as follows:

1. I agree that the inclusion of ageism as a form of EAV could lead to inflated rates of EAV, at least in terms of how it is typically conceptualized. Please elaborate on the motivation for its inclusion and if possible, provide the number of articles included in the review that were based only on the link between ACEs and ageism in populations 60 and over. If one or more were included based on only this specific association, please discuss the implications of including it on the overall findings from the review.

2. Inclusion and exclusion criteria should offer more details, as noted by the Reviewer.

3. Please address the Reviewer's question about whether inclusion required a finding of a statistically significant relationship.

4. Please review details offered about specific studies to ensure accurate reporting.

5. Please clarify what is meant by cumulative EAV.

6. Please review the conclusion section after making the revisions outlined above and add other relevant takeaways, if warranted. 

We look forward to receiving your revised manuscript.

Kind regards,

Kristen Slack

Academic Editor

PLOS ONE

Additional Editor Comments:

Thank you for the opportunity to review your submission. I encourage you to revise and resubmit your manuscript by following the guidance below for minor revisions.

Please address Reviewer 1 numbered comments as follows:

1. Provide a more concise overview of the findings in the abstract.

2. Provide a citation supporting the claim of a higher prevalence of EAV in low- and middle-income countries compared to higher income countries.

3. The sentence in question reads as if there is only one pathway from ACEs to EAV, which is not true. Please revise this sentence to reflect that the pathway described is one of several potential pathways, of particular relevance to the present study, and provide relevant citations.

4. I believe this paragraph is relevant. However, please shorten and focus on other systematic reviews linking ACEs to other outcomes and to the highlighted claim regarding the absence of systematic reviews on ACEs and EAV.

5. Please be sure to temper the claim that there has been "a significant number of bodies of research on this association". The outcome of the systematic review itself suggests that this is not the case and contradicts a later claim that there is a "low volume" of scholarship on this topic in research (Critical finding, page 20).

6. Please thoroughly review the submission for typos.

7. The criteria are met for a systematic review.

8. Please expand discussion on measurement inconsistencies and use more caution in the discussion about moderating factors, as described by the Reviewer.

9. Please expand the discussion on the justification for applying a life course perspective, as described by the Reviewer.

10. Be sure to address the comments of Reviewer 1 in terms of what to emphasize in more detail. I would also like to see a brief mention of possible limitations of excluding grey literature from the search process.

Please address Reviewer 2 comments as follows:

1. I agree that the inclusion of ageism as a form of EAV could lead to inflated rates of EAV, at least in terms of how it is typically conceptualized. Please elaborate on the motivation for its inclusion and if possible, provide the number of articles included in the review that were based only on the link between ACEs and ageism in populations 60 and over. If one or more were included based on only this specific association, please discuss the implications of including it on the overall findings from the review.

2. Inclusion and exclusion criteria should offer more details, as noted by the Reviewer.

3. Please address the Reviewer's question about whether inclusion required a finding of a statistically significant relationship.

4. Please review details offered about specific studies to ensure accurate reporting.

5. Please clarify what is meant by cumulative EAV.

6. Please review the conclusion section and add more details about the findings and their relative strengths, given the small number of studies to date.

Reviewers' comments:

Reviewer's Responses to Questions

**Comments to the Author**

1. Is the manuscript technically sound, and do the data support the conclusions?

Reviewer #1: Yes

Reviewer #2: Partly

2. Has the statistical analysis been performed appropriately and rigorously? 

Reviewer #1: Yes

Reviewer #2: Yes

3. Have the authors made all data underlying the findings in their manuscript fully available?

Reviewer #1: Yes

Reviewer #2: Yes

4. Is the manuscript presented in an intelligible fashion and written in standard English?

Reviewer #1: Yes

Reviewer #2: Yes

5. Review Comments to the Author

Reviewer #1: Thanks for the opportunity to review the manuscript titled Adverse Childhood Experiences and Elder Abuse Victimization Nexus: A systematic review and meta-analysis. Please find my comments below:

1. The abstract is long and the report of the results was hard to follow. I recommend a concise and thorough summary of the results in the abstract.

2. On Page 4: it is stated that “… the prevalence of abuse in many countries, especially low- and middle-income countries is expected to increase with the increasing rate of population ageing (WHO, 2022)”.

It is unclear why the issue is especially relevant for low- and middle-income countries? There is no clear explanation even after checking the citation.

3. The sentences related to ‘one pathway’ on page 5 do not have any in-text citations. Also, this section should be discussed more thoroughly based on the review of prior studies.

4. The paragraph about the ‘associations between ACEs and adverse outcomes’ on page 5 does not seem relevant to this study.

5. I disagree with the sentence: “following an emerging recognition of the damaging impact of ACEs on EAV in later life, there has been a significant number of bodies of research on this association.” This is simply not accurate, as this systematic review included only 9 studies. The authors need to provide a better rationale and significance for this study. Given the small number of existing studies and the emerging nature of this topic, I am not sure if this systematic review would have been necessary.

6. There were typos throughout the introduction section.

7. The methodology of this paper appears to be thorough, but I wonder how the small number of papers (n = 9) may affect the pooled results.

8. In discussion, I recommend expanding the discussion about measurement consistencies: how they are inconsistent and what are the problems associated with the inconsistent measurement. Also, the discussion about moderating factors should be approached carefully since they were based on two studies only.

9. On page 19: I recommend expanding the discussion about ‘the association between ACEs and EAV aligns with the life course perspective’. How does it align and are there other theories that can explain the association? Authors can consider bringing in the prior literature about lifetime revictimization.

10. The discussion about the strengths and limitations of the study was very brief, which needs to be strengthened and clearly specify the contribution of this study.

Reviewer #2: 1. The authors state the following as it relates to forms of EAV: “Examples of this victimisation include older adults’ experience of ageism, financial exploitation, sexual abuse, physical and psychological assault or abuse, and neglect (Dong, 2015).” Ageism is generally not regarded as a form of EAV, but rather as a risk factor. Including ageism as a form of EAV would indicate a prevalence of 100% among older adults, since all older adults are exposed to ageism, albeit in different ways based on identity.

2. As it relates to study eligibility for inclusion in the review, did you consider qualitative versus quantitative studies, year of publication, location of study, or type of analysis that examined the relationship between ACEs and EAV?

3. To be included in the review, did the study need to have reported a significant association between ACEs and EAV, or did it also include studies that found no significant relationship?

4. The authors are encouraged to review the accuracy of information captured on studies included in the review. For example, the Burnes et al (2022) study did not use a three items questionnaire to measure EAV, nor were there 36 items included in the child maltreatment questionnaire.

5. What is meant by cumulative EAV? Do the authors mean aggregate or total EAV that includes any subtype?

6. The following conclusion reported in the discussion seems too strong, given that the respective results were based on only one study: “Our results further show that gender, physical health, and depressive symptoms moderate the association between ACE and EAV.”

6. PLOS authors have the option to publish the peer review history of their article (what does this mean?). If published, this will include your full peer review and any attached files.

Reviewer #1: No

Reviewer #2: No

---

## [Author Response · Author response to Decision Letter 0]

27 Sep 2024

Dear Kristen

Thank you very much for the opportunity to respond to the reviewer’s comments. We want to emphasize that the suggestions made by the reviewers have improved the manuscript. We have addressed the reviewer’s comments below: 

Response to Reviewer’s comments

Please address Reviewer 1 numbered comments as follows:

1. Provide a more concise overview of the findings in the abstract.

Response: 

A more concise abstract has been provided. 

2. On Page 4: it is stated that “… the prevalence of abuse in many countries, especially low- and middle-income countries is expected to increase with the increasing rate of population ageing (WHO, 2022)”. It is unclear why the issue is especially relevant for low- and middle-income countries? There is no clear explanation even after checking the citation.

Response: 

Thank you for this comment. I have modified the sentence and included another reference and it now reads as “In addition to the reported increase of abuse during the Coronavirus disease – 2019 (COVID-19) (Chang, 2016), the prevalence of abuse in many countries, including low- and middle-income countries is expected to increase with the increasing rate of population ageing (World Health Organization, 2024; Yon et al., 2017).” The article published by WHO in 2022 was based on one of the studies which is a systematic review and meta-analysis, reporting a high prevalence of elder abuse in low- and middle-income countries as well. Out of the 28 studies included in the systematic review, 7 were based in low- and middle-income countries. 

3. Provide a citation supporting the claim of a higher prevalence of EAV in low- and middle-income countries compared to higher income countries.

Response: 

Thank you. The two citations (World Health Organisation (2024) and Yon et al., (2017) have been provided. 

4. The sentence in question reads as if there is only one pathway from ACEs to EAV, which is not true. Please revise this sentence to reflect that the pathway described is one of several potential pathways, of particular relevance to the present study, and provide relevant citations. 

Response: 

The sentence has been revised and referenced. It now reads as “One of these pathways explored in this study is that older adults who experience adverse circumstances in their childhood, living in any setting, may carry disempowerment with them throughout later life, and are more likely to be re-victimized in the community or when living in a residential setting (Asyraf et al., 2021)”

5. I believe this paragraph is relevant. However, please shorten and focus on other systematic reviews linking ACEs to other outcomes and to the highlighted claim regarding the absence of systematic reviews on ACEs and EAV.

Response: 

Thanks for this suggestion. I have made the argument more concise. It now reads as “For instance, systematic reviews have provided evidence on ACEs, or effects of multiple ACEs on health outcomes such as sexual, mental, and physical health, and higher utilisation of healthcare (Hughes et al., 2017; Kalmakis & Chandler, 2015; Petruccelli et al., 2019)

6. Please be sure to temper the claim that there has been "a significant number of bodies of research on this association". The outcome of the systematic review itself suggests that this is not the case and contradicts a later claim that there is a "low volume" of scholarship on this topic in research (Critical finding, page 20).

Response: 

We have acknowledged this discordance and addressed it accordingly. The claim now reads as “Moreover, following an emerging recognition of the impact of ACEs on EAV in later life, there have been some papers published on the association (Burnes et al., 2022; Easton & Kong, 2021; McDonald & Thomas, 2013; Wiklund et al., 2022)

7. Please thoroughly review the submission for typos.

Response

We have read the entire manuscript for typo corrections. 

7. The criteria are met for a systematic review.

Response: 

Thank you very much for this comment. We appreciate this feedback. 

8. In discussion, I recommend expanding the discussion about measurement consistencies: how they are inconsistent and what are the problems associated with the inconsistent measurement. Also, the discussion about moderating factors should be approached carefully since they were based on two studies only

Response: 

We have expanded the discussion on measurement and their potential impacts. Also, we have modified our discussion on the moderating factors elaborating on the possible implications for research, policy and practice. 

9. Please expand the discussion on the justification for applying a life course perspective, as described by the Reviewer. On page 19: I recommend expanding the discussion about ‘the association between ACEs and EAV aligns with the life course perspective’. How does it align and are there other theories that can explain the association? Authors can consider bringing in the prior literature about lifetime revictimization. The discussion about the strengths and limitations of the study was very brief, which needs to be strengthened and clearly specify the contribution of this study.

Response: 

Thank you for the suggestions to elaborate on the justification for applying the life course perspective. We have provided relevant information including the justification for the life course perspective in the introduction making it easier for the reader to understand the discussion of the life course perspective in the discussion section. 

10. Be sure to address the comments of Reviewer 1 in terms of what to emphasize in more detail. I would also like to see a brief mention of possible limitations of excluding grey literature from the search process. 

Response:

Thank you for these comments. I have included in the discussion that, “Moreover, though there was no study excluded based on grey literature status, the inclusion of this as an exclusion criterion might have automatically excluded grey literature that reported on the association between ACEs and EAV in the first place.”

Please address Reviewer 2 comments as follows:

I agree that the inclusion of ageism as a form of EAV could lead to inflated rates of EAV, at least in terms of how it is typically conceptualized. Please elaborate on the motivation for its inclusion and if possible, provide the number of articles included in the review that were based only on the link between ACEs and ageism in populations 60 and over. If one or more were included based on only this specific association, please discuss the implications of including it on the overall findings from the review.

Response: 

Thank your for your comment on EAV and ageism. Ageism is a bias against older adults groups based on their age, which may take the form of discrimination at all levels against such individuals and groups, up to and including victimisation and bullying (Pillemer, Burnes, & MacNeil, 2021). Investigating the connection between ageism and elder mistreatment. Nature Aging, 1(2), 159-164.). In view of this, we included ageism as a synonym for elder abuse victimisation. We want to profoundly say that none of the studies included specifically measured the association between ACEs and ageism. 

2. Inclusion and exclusion criteria should offer more details, as noted by the Reviewer. As it relates to study eligibility for inclusion in the review, did you consider qualitative versus quantitative studies, year of publication, location of study, or type of analysis that examined the relationship between ACEs and EAV?

Response

We have provided more information on the inclusion strategies. Under the “study eligibility criteria”, we have included that the Inclusion of studies was not restricted to any research approach, year of publication, location of study, or type of analytical method used to examine the association on ACEs and EAV. These inclusion strategies allowed for any study available on the ACEs – EAV association to be included in the systematic review.

3. Please address the Reviewer's question about whether inclusion required a finding of a statistically significant relationship. To be included in the review, did the study need to have reported a significant association between ACEs and EAV, or did it also include studies that found no significant relationship?

Response: 

The significance level as reported by the study did not determine its inclusion in the systematic review. The most relevant inclusion indicator was for the study to report examining as association between ACEs and EAV. 

4. Please review the details offered about specific studies to ensure accurate reporting. The authors are encouraged to review the accuracy of information captured on studies included in the review. For example, the Burnes et al (2022) study did not use a three items questionnaire to measure EAV, nor were there 36 items included in the child maltreatment questionnaire.

Response: 

We have reviewed the included studies for accurate reporting. 

5. Please clarify what is meant by cumulative EAV.

Response: 

We have replaced the cumulative EAV with ‘multiple’ to represent an experience of more than one elder abuse victimisation. This could include experiencing say emotional abuse and physical abuse together, or emotional abuse, physical abuse and financial abuse. 

6. Please review the conclusion section after making the revisions outlined above and add other relevant takeaways, if warranted. 

Response: 

Thank you very much. We have reviewed the conclusion of the review to capture all important take home messages.

---

## [Editor Report · Decision Letter 1]

30 Oct 2024

PONE-D-24-07611R1Adverse childhood experiences and elder abuse victimization nexus: A systematic review and meta-analysisPLOS ONE

Dear Dr. Awuviry-Newton,

Thank you for submitting your manuscript to PLOS ONE. After careful consideration, we feel that it has merit but does not fully meet PLOS ONE’s publication criteria as it currently stands. Therefore, we invite you to submit a revised version of the manuscript that addresses the points raised during the review process.

We look forward to receiving your revised manuscript.

Kind regards,

Kristen Slack

Academic Editor

PLOS ONE

Journal Requirements:

Additional Editor Comments:

This is a strong paper that makes an important contribution to the research literature. While you sufficiently addressed reviewers’ concerns in this revision, a thorough editing of the manuscript for grammar and syntax was not apparent. PLOS ONE does not provide on-staff editorial assistance, so the job of an editorial board member includes reviewing manuscripts for grammar, punctuation, syntax, and clarity consistent with the journal’s requirements. In future submissions, please take this into account as it will significantly speed up the review process. While I have many comments, they are all minor and require a wording change or attention to the use of singular vs. plural nouns, use of commas, and the like. I have uploaded my Word document with my editorial  changes and additional comments to make your review of my changes easier to work with, rather than listing the needed changes here.

---

## [Author Response · Author response to Decision Letter 1]

4 Nov 2024

Responses to Comments

Dear Editor, 

We thank you very much for the comments raised, which have improved the paper immensely. We have addressed the comments in the manuscript and also below; 

Journal Requirements:

Response

I have reviewed the reference list for accuracy. The reference is complete and correct. 

Additional Editor Comments:

This is a strong paper that makes an important contribution to the research literature. While you sufficiently addressed reviewers’ concerns in this revision, a thorough editing of the manuscript for grammar and syntax was not apparent. PLOS ONE does not provide on-staff editorial assistance, so the job of an editorial board member includes reviewing manuscripts for grammar, punctuation, syntax, and clarity consistent with the journal’s requirements. In future submissions, please take this into account as it will significantly speed up the review process. While I have many comments, they are all minor and require a wording change or attention to the use of singular vs. plural nouns, use of commas, and the like. I have uploaded my Word document with my editorial changes and additional comments to make your review of my changes easier to work

Response

Thank you very much for the thorough proofreading of the manuscript. We have addressed all comments raised in the manuscript. 

General comments:

1. Use “in singular or multiple form” rather than “by form or multiple” throughout manuscript. Example from the abstract: “All studies that addressed associations between ACEs, by form or multiple in singular or multiple form, and EAV were included in the review.” I note that you did attend to this prior comment from me in this revision, but in some instances in the manuscript, the word “cumulative” still appears. I note that measures capturing multiple forms of ACEs or EAVs can be measured dichotomously (e.g., having multiple vs. none or having particular combinations), and not necessarily cumulatively (i.e. a sum or count), so use these two terms with intentionality. For example, when I see the word “cumulative”, I expect the model coefficient for a predictor to represent the predicted change in an association for each additional number of something (e.g., ACEs) included in a count variable. It may be that you have intentionally chosen to use one of these terms over the other—I am just requesting that you double-check and where appropriate, use consistent terminology.

Response

I have perused the entire manuscript and now consistent with the use of “in singular or multiple form”. 

2. Once you spell out “adverse childhood experiences” within the main text following the abstract, and give the acronym “(ACEs)”, you can use ACE throughout the manuscript. If you choose to use the fully spelled out term interchangeably throughout, then, at a minimum, be consistent in subsection titles (use “ACEs” or use “adverse childhood experiences”).

Response

This has been taken care of in the entire manuscript. 

3. Check that singular vs. plural is used correctly (e.g., in the abstract, “odds ratios (OR)” is plural but “confidence interval (CI) is singular. When referring to one as plural, the other should be plural.

Response

I have corrected this in the abstract and in the manuscript as well. 

4. Check appropriate use of commas throughout manuscript. Some, but not all, examples are given below.

When you refer to EAV, you sometimes use EAVs. It would sound strange to say “elder abuse victimizations” if you are simply referring to any form of EAV. Please check throughout the document to make sure you intend to use a plural form of EAV.

Response

I have corrected this problem in the entire manuscript. 

5. Check for unnecessary double-spacing within sentences.

Response

I have removed all unnecessary spaces within the manuscript. 

6. The comments above should be addressed throughout the manuscript. In some comments below, I have identified where these types of changes should be made, but not all instances were flagged.

Abstract:

Add an “s” to “association”: Included studies examined multiple ACEs and multiple EAV associations (N=3)

Awkwardly worded sentence: The results indicate that interventions designed to reduce ACEs, by form or multiple in singular or multiple form, early in life and to support residential-and community-dwelling older adults may be relevant in reducing the incidence of EAV

“The life course perspective should be integrated….”

Response

I have addressed all of these comments raised here in the manuscript. 

Background:

(Page 3)

7. Make “pose” plural: “The growing prevalence and incidence of elder abuse victimization (EAV) reported in residential and community (home care) settings across the world poses both a public health and a human rights concern.”

Add comma after second use of the word “countries”: “…the prevalence of abuse in many countries, including low- and middle-income countries, is expected to increase…”

Change wording: “Increasing levels of elder abuse co-occur with high levels of child maltreatment and adverse childhood experiences (ACEs) (Madigan et al., 2023).”

Response

We have rectified the above issue in the manuscript. 

8. Since the word “adults’ “ is a possessive plural, delete “the” and make “experience” plural: “EAV, in this review, will refer to the older adults’ experiences of a single or repeated act or lack of appropriate action…”

Add comma after “knowledge”, change “enunciate” to “elucidate”: “However, to our knowledge, no systematic syntheses or reviews currently exist to enunciate explore the nature of this connection/pathway”.

(Page 4)

“Specific examples of ACEs include childhood sexual abuse, parental criminality and divorce , childhood disempowerment, and childhood discrimination (Hughes et al., 2017).”

What does “This” refer to?: “This includes the cumulative harmful effects of at least two domains of ACEs (Hughes et al., 2017).”

Response: 

We have addressed all the above comments in the manuscript. 

9. In the following sentence, change “these” to “the” since you are referring to the singular noun “One”: “One of these pathways explored in this study…”

Change wording: “There has been research Studies on the associations between ACEs and adverse outcomes, with use varying vary in their designs, methodologies, locations, findings and conclusions.”

Drop the “s” in “influences”: “For example, a study conducted in China revealed that social and economic instability during childhood influences an increase in…”

Add comma after second “ACEs”: “For instance, systematic reviews have provided evidence on ACEs, or effects of multiple ACEs, on health outcomes such as…”

Simplify sentence: “Moreover, following an emerging recognition of the impact of ACEs on EAV in later life, there have been some In addition to an earlier study (McDonald & Thomas, 2013), more recent research has focused on the association between ACEs and EAV (Burnes et al., 2022; Easton & Kong, 2021; McDonald & Thomas, 2013; Wiklund et al., 2022).

Response

The above comments have been addressed in the manuscript. 

(Page 5):

10. Change wording: “Synthesizing evidence on this association will be significant in contribute to understanding, or revealing, a clear pathway of the association and evidence on how to mitigate this public health concern.”

Change wording: “In this study, As applied elsewhere (McDonald & Thomas, 2013), we employed the life-course perspective as a useful framework to understand for interpreting the findings of the systematic review as applied elsewhere (McDonald & Thomas, 2013)…”

Use plural for “experience”: “…about aspects of developmental experiences…”

Add “a” or “the”: “According to Elder (2006), a life-course perspective elucidates the potential effects of accumulative advantage or disadvantage of individuals – social structure relationship on individuals over time. ”

Drop this sentence or elaborate on this example and provide a citation: “To illustrate this, the experience of a child living in an institution such as a school or a family setting will continue to be important in later life.”

Combine sentences: “Accordingly, This systematic review aimed to understand the connections between ACEs and EAVs, as well as . We also need to gain insight into factors moderating the relationships between ACEs – EAVs relationships.

Change wording: “Understanding An enhanced understanding of this evidence using a life-course perspective would assist in the development of appropriate interventions for older adults who experience abuse , prevent the occurrence of ACEsthrough the understanding of life-course perspectives, and provide an urgent and clear a clearer pathway for more improve cross-cultural and longitudinal research on the relationships between ACEs and EAV.”

Response

All the comments raised above have been addressed in the manuscript. 

(Page 6):

11. Change wording: “Two authors (KAN and KAC) did the screening independently and met on Zoom to address any disagreement for and achieve consensus.”

Change wording: “Studies on elder abuse victimization, or aspect on elder abuse types , such as ageism, psychological abuse,…”

Change wording: “Inclusion of studies were Study inclusion was not restricted…”

Change wording; add comma after “manager”, remove comma after “removal”: “Articles that resulted from the identified in databases searched were exported to Endnote, a reference manager, to allow for duplicate removal, and title and abstracts screening, as per the study’s inclusion criteria.”

Response

All comments above have been addressed in the manuscript. 

(Page 6-7):

Remove comma after “retrieved”; change wording: All articles that potentially met eligibility criteria were retrieved, after the a full test screen of all potential articles was conducted. Afterwards After consulting with the other three authors, all articles meeting the eligibility criteria were retained. After consulting with the other three authors.

Response

We have addressed the above comments. 

(Page 7):

12. Add comma after “August 2023”: “The search was conducted in English and performed in PubMed, CINAHL Complete, APA PsycINFO, Web of Science, Scopus, and Google Scholar in August 2023, covering articles published in or before 2023.”

Respone

We have addressed the above comment in the manuscript. 

(Page 8):

13. Add comma before “respectively”: The checklists were made up of 8 and 11 questions for analytical cross-sectional and cohort studies, respectively.

Change wording: “After eligible studies had been included were identified,…”

Should “structures” be singular?: “This grouping generated themes providing structures for the presentation of the findings.”

Response

We have addressed comments above in the manuscript. 

(Page 9):

14. Make “interval” plural”: After the narrative review was completed for a particular theme, a meta-analysis of the extracted odds ratios and confidence intervals.”

Add comma after ratios and drop the word “and”: “Pooled odds ratios, and confidence intervals and heterogeneity were used to present consolidated associations for the themes identified, where applicable.”

Change wording: “After title and abstract screening had been was completed…”

Change wording: “The full text screening resulted in a total of 9 articles (all employed quantitative methods) that met eligibility criteria for the systematic review…”

Change wording: “None was were conducted in an African country.”

Response

The above comments have been addressed in the manuscript. 

(Page 10):

15. Change wording and address plural/singular and tense corrections: “The sample sizes of the included studies rangeds from 135 to 23,468, with the and participants were being predominantly females in all included studies.”

“Participants in the included studies were aged 60 years and above, who reported ACEs and EAV experiences in later life.” 

Change wording: “Some of the studies measured ACEs cumulatively on cumulative (Asyraf et al., 2021; Chen & Fu, 2022b; Dong & Wang, 2019; Easton & Kong, 2021) whiles others used a measured types of ACE separately (Burnes et al., 2022; Giraldo-Rodríguez et al., 2022; Kong & Easton, 2019; McDonald & Thomas, 2013; Wiklund et al., 2022) to measure childhood experience.”

Change wording: “The studies reported using different types of measurement tools with unequal and unique sets of questions to measure ACEs.”

Change wording: “….and a questionnaire measuring an abuse before the age of 65. Years questionnaire

Delete sentence as it is redundant with first sentence in the paragraph: “All of these questionnaires were used to measure ACEs either on cumulative or in part.”

Response

We have addressed all the comments above in the manuscript. 

(Page 11):

Change wording: “Even though cumulative EAV (cumulative) was the main the most common outcome measure among the studies (Asyraf et al., 2021; Burnes et al., 2022; Chen & Fu, 2022b; Easton & Kong, 2021; Kong & Easton, 2019), other single forms of EAV outcomes contributing to our understanding of EAV were reported, as well, including These single outcome measures included experience of” physical elder abuse…”

Change wording: “The included studies used distinct scales and question items to measure EAV, including The included studies used sets of questions including the Abusive Behavior Inventory…”

Check official name of measure, as I’ve only seen it referenced with “scales” capitalized and singular: “…Revised Conflict Tactics scales (Asyraf et al., 2021)…”

Change wording in section title: Is it “multiple” or cumulative”?—be sure you are using these terms purposively. “Multiple” forms could be operationalized in a non-cumulative way (e.g., a dichotomous variable). If you always mean cumulative, then section title should use that word instead of “multiple”; either capitalize each separate word in ACEs or use small case for “adverse” in section title; also, “experience” should be plural, correct?: “Connections between Adverse childhood experiences and multiple elder abuse victimization”

Change wording and check whether “multiple” should be “cumulative” and the reverse: ” Seven studies examined the association between ACEs (multiple or single forms) or part and cumulative EAV.”

Response

The comments have been addressed in the manuscript. 

(Page 12):

Delete “in”: “Burnes et al. (2022) and Dong and Wang (2019) reported that a history, or increasing incidence of in child maltreatment…”

Include direction (e.g., “positive” association) of the relationship: “Similarly, ACEs had a significant association with EAV (Easton & Kong, 2021).”

Change wording; also, should “ACE” be plural here? If not, change to “at least one ACE and”: “The pooled odds ratio from a meta-analysis shows a statistically significant association between experiencing ACEs and multiple EAV…”

Change wording: “Some of the studies examined the effect of having a history of a co-occurring abuse/ACEs on EAV in later life. While most of the results were consistent, some contradictions were apparent. Experiencing either three or at least four ACEs was associated with EAV (Asyraf et al., 2021; Chen & Fu, 2022b; Giraldo-Rodríguez et al., 2022). While a significant association with EAV was also suggested when a child experienced one or two ACEs (Giraldo-Rodríguez et al., 2022), Asyraf et al. (2021) experienced found no significant association.

Clarify the implication of this statement: “Despite this, heterogeneity existed across studies…”

Response

The above comments have been addressed in the manuscript. 

(Page 13):

Use “While” not “Whiles”: “Whiles in the Mexican study….”

Consider simply saying the following: “In both a similar geographic contexts (Mexico and the 

---

## [Decision Letter · Decision Letter 2]

18 Dec 2024

Adverse childhood experiences and elder abuse victimization nexus: A systematic review and meta-analysis

PONE-D-24-07611R2

Dear Dr. Awuviry-Newton,

We’re pleased to inform you that your manuscript has been judged scientifically suitable for publication and will be formally accepted for publication once it meets all outstanding technical requirements.

Kind regards,

Rafi Amir-ud-Din

Academic Editor

PLOS ONE

Additional Editor Comments (optional):

Reviewers' comments:

Reviewer's Responses to Questions

**Comments to the Author**

1. If the authors have adequately addressed your comments raised in a previous round of review and you feel that this manuscript is now acceptable for publication, you may indicate that here to bypass the “Comments to the Author” section, enter your conflict of interest statement in the “Confidential to Editor” section, and submit your "Accept" recommendation.

Reviewer #1: All comments have been addressed

Reviewer #2: All comments have been addressed

2. Is the manuscript technically sound, and do the data support the conclusions?

Reviewer #1: Yes

Reviewer #2: Yes

3. Has the statistical analysis been performed appropriately and rigorously? 

Reviewer #1: Yes

Reviewer #2: Yes

4. Have the authors made all data underlying the findings in their manuscript fully available?

Reviewer #1: Yes

Reviewer #2: Yes

5. Is the manuscript presented in an intelligible fashion and written in standard English?

Reviewer #1: Yes

Reviewer #2: Yes

6. Review Comments to the Author

Reviewer #1: The authors did an excellent job addressing the reviewers' concerns. I am content with their revision.

Reviewer #2: (No Response)

7. PLOS authors have the option to publish the peer review history of their article (what does this mean?). If published, this will include your full peer review and any attached files.

Reviewer #1: No

Reviewer #2: No

---

## [Editor Report · Acceptance letter]

17 Jan 2025

PONE-D-24-07611R2 

PLOS ONE

Dear Dr. Awuviry-Newton, 

I'm pleased to inform you that your manuscript has been deemed suitable for publication in PLOS ONE. Congratulations! Your manuscript is now being handed over to our production team.

Kind regards, 

on behalf of

Dr. Rafi Amir-ud-Din 

Academic Editor

PLOS ONE